# Detection of Four Porcine Enteric Coronaviruses Using CRISPR-Cas12a Combined with Multiplex Reverse Transcriptase Loop-Mediated Isothermal Amplification Assay

**DOI:** 10.3390/v14040833

**Published:** 2022-04-17

**Authors:** Jiajia Liu, Dagang Tao, Xinquan Chen, Linyuan Shen, Li Zhu, Bingrong Xu, Hailong Liu, Shuhong Zhao, Xinyun Li, Xiangdong Liu, Shengsong Xie, Lili Niu

**Affiliations:** 1College of Animal Science and Technology, Sichuan Agricultural University, Chengdu 611130, China; jjliu0419@foxmail.com (J.L.); chenxinquan0705@foxmail.com (X.C.); shenlinyuan0815@163.com (L.S.); zhuli@sicau.edu.cn (L.Z.); 2Key Laboratory of Agricultural Animal Genetics, Breeding and Reproduction of Ministry of Education and Key Lab of Swine Genetics and Breeding of Ministry of Agriculture and Rural Affairs, Huazhong Agricultural University, Wuhan 430070, China; dagangtao@foxmail.com (D.T.); xubingrongxu@163.com (B.X.); hailongliu@webmail.hzau.edu.cn (H.L.); shzhao@mail.hzau.edu.cn (S.Z.); xyli@mail.hzau.edu.cn (X.L.); liuxiangdong@mail.hzau.edu.cn (X.L.); 3The Cooperative Innovation Center for Sustainable Pig Production, Huazhong Agricultural University, Wuhan 430070, China; 4Hubei Hongshan Laboratory, Frontiers Science Center for Animal Breeding and Sustainable Production, Wuhan 430070, China

**Keywords:** porcine enteric coronaviruses, CRISPR/Cas12a, multiplex loop-mediated isothermal amplification, single-strand DNA-FQ reporter

## Abstract

Porcine enteric coronaviruses have caused immense economic losses to the global pig industry, and pose a potential risk for cross-species transmission. The clinical symptoms of the porcine enteric coronaviruses (CoVs) are similar, making it difficult to distinguish between the specific pathogens by symptoms alone. Here, a multiplex nucleic acid detection platform based on CRISPR/Cas12a and multiplex reverse transcriptase loop-mediated isothermal amplification (RT-LAMP) was developed for the detection of four diarrhea CoVs: porcine epidemic diarrhea virus (PEDV), transmissible gastroenteritis virus (TGEV), porcine deltacoronavirus (PDCoV), and swine acute diarrhea syndrome coronavirus (SADS-CoV). With this strategy, we realized a visual colorimetric readout visible to the naked eye without specialized instrumentation by using a ROX-labeled single-stranded DNA-fluorescence-quenched (ssDNA-FQ) reporter. Our method achieved single-copy sensitivity with no cross-reactivity in the identification and detection of the target viruses. In addition, we successfully detected these four enteric CoVs from RNA of clinical samples. Thus, we established a rapid, sensitive, and on-site multiplex molecular differential diagnosis technology for porcine enteric CoVs.

## 1. Introduction

Coronaviruses (CoVs) are a class of viruses that cause respiratory tract infections in mammals and birds. The porcine enteric CoV has spread on a large scale worldwide, severely affecting the efficiency of pork production and the animal husbandry economy [1,2,3]. It is currently known that four porcine CoVs can cause porcine diarrhea, including three α-CoVs (transmissible gastroenteritis virus (TGEV), porcine epidemic diarrhea virus (PEDV), and swine acute diarrhea syndrome coronavirus (SADS-CoV), as well as one δ-CoV [Porcine Deltacoronavirus (PDCoV)] [4,5,6]. Among them, TGEV has been circulating in pigs for decades, whereas PEDV, PDCoV, and SADS-CoV are considered as emerging CoVs [6]. Moreover, the latest research has shown that SADS-CoV replicates effectively in human cells, suggesting that the virus is a potential threat to humans [7]. However, these four porcine enteric CoVs cannot be distinguished by their clinical symptoms [8]. Therefore, the development of low-cost, innovative point-of-care (POC) tests for detection and diagnosis of porcine CoVs has become a priority in biosensors and bioassays.

A variety of porcine CoVs detection methods are currently applied, including immunofluorescence (IF) or immunohistochemistry (IHC) tests, in situ hybridization, nucleic acid amplification-based methods, etc. Among them, Reverse Transcription-Polymerase Chain Reaction/Real-time quantitative (RT–PCR/qPCR) is the main laboratory diagnosis method for detecting porcine CoVs, mainly targeting conservative genes, such as *M*, *N*, or *ORF3* genes [9,10,11,12,13]. The multiplex PCR detection method for viruses is fast and efficient and has been developed in the nucleic acid detection field [12]. This assay for the detection of major diarrheal viruses in pig herds has greatly improved the efficiency of detection [12,14,15]. However, there are still many limitations in PCR-based diagnoses, such as expensive equipment, a lack of well-trained operators, and limited laboratory environments, which pose many challenges in conducting diagnoses outside the laboratory [16,17]. Therefore, it is urgent to develop a POC diagnostic platform for rapidly detecting multiple porcine diarrhea CoVs in the field, which can be quickly applied to diagnosis in resource-poor or remote small pig farms. Especially for the simultaneous on-site measurement of different CoVs from a single sample, multiplexed POC testing is necessary.

Recently, clustered regularly interspaced short palindromic repeat (CRISPR)- and CRISPR-associated protein (Cas)-based biosensing platforms have been developed for their high specificity, simplicity, and ease of field deployment. CRISPR-associated endoribonucleases, such as Cas13a, Cas12a, Cas12b, and Cas14, perform indiscriminate collateral cleavage activities when the Cas/small guide RNA (sgRNA)/target ternary complex is formed. By combining these Cas proteins with a preamplification step, such as PCR, recombinase polymerase amplification (RPA) or loop-mediated isothermal amplification (LAMP) and single-stranded DNA-fluorescence-quenched (ssDNA-FQ) reporter, researchers have developed various biosensing platforms for nucleic acid detection: SHERLOCK [18], DETECTR [19], HOLMES [20], CDetection [21] and Cas14-DETECTR [22]. These provide attomolar sensitivity and single-base specificity for the one-site detection of pathogenic viruses [23,24,25], such as severe acute respiratory syndrome (SARS)-CoV-2 [26,27] and African swine fever virus (ASFV) [28,29]. However, these CRISPR-based nucleic acid detection technologies still require blue or UV light transilluminators for fluorescence readout and expensive commercial lateral flow strips for visual readout.

In this study, we developed a CRISPR-/Cas12a-based nucleic acid colorimetric detection system combined with multiplex LAMP (multiplex LAMP-Cas12a), which allows the detection of PEDV, TGEV, PDCoV, and SADS-CoV with the naked eye. We also evaluated its application in RNA of clinical samples collected from infected farm pigs. Thus, our technique can be used to rapidly and efficiently detect the four pathogenic swine enteric CoVs.

## 2. Materials and Methods

### 2.1. Preparation of Nucleic Acid Template

The target *ORF3* gene of virulent field PEDV strain (KJ642645.1), the *N* gene of TGEV (GQ374566.1), the *N* gene of PDCoV (KY586149.1), and the *N* gene of SADS-CoV (MG775253.1) and their corresponding partial fragments were synthesized (TSINGKE, Beijing, China) according to the highly conserved region of the gene sequences in GenBank (https://www.ncbi.nlm.nih.gov/genbank/, accessed on 20 February 2022) and cloned into the pMD18T vector (Appendix A).

The RNAs of clinical samples of PEDV, TGEV, and PDCoV were extracted from the supernatant of virus-infected cells or tissues samples. SADS-CoV RNA was transcribed in vitro. Specifically, the SADS-CoV-N plasmid was amplified using a forward primer containing a T7 promoter and a reverse primer (Appendix A). The volume of the PCR assay was 20 μL, composed of 10 μL of Extaq Mix, the prime mixture (10 pmol forward primer and 10 pmol reverse primer), and 5 ng of the DNA template. The PCR product was purified using a PCR Purification Kit (Beijing, China, Tianmo Sci&Tech Development). The amplified purified PCR product then served as the DNA template for the in vitro transcription reactions using the HiScribe™ T7 High Yield RNA Synthesis Kit (E2040S, NEB, Ipswich, MA, USA) following the manufacturer’s instructions. The in vitro transcribed RNA was purified by acid-phenol-chloroform (NEB) extraction and resuspended in RNase-free water following isopropanol precipitation. The concentration of the RNA was quantified by a NanoDrop One™ spectrophotometer (Software version 2.7, Thermo Fisher Scientific, Waltham, MA, USA).

### 2.2. Preparation of sgRNAs

SgRNAs targeting conserved genes were designed with CRISPR-offinder software [30]. The designed sgRNA was complemented to the target sites with a “5′TTTN” protospacer adjacent motif (PAM) in the DNA strand opposite the target sequences (Appendix A). The sgRNA templates were amplified from a pUC57-T7-sgRNA (Appendix A) using a forward primer containing a T7 promoter and a reverse primer containing different nucleotide target sequences. Then, sgRNAs were transcribed and purified in vitro.

### 2.3. CRISPR/Cas12a Combined with PCR Assay

Primers were designed by Primer Premier 5.0 (PREMIER Biosoft) based on the highly conserved sequences of the *ORF3* gene targeting PEDV, which were synthesized and cloned into the pMD18T vector (Appendix A). PCR assay was performed in a thermal cycler PCR system with an amplification reaction as follows: 30 cycles of denaturation at 94 °C for 30 s, annealing at 55 °C for 30 s, extension at 72 °C for 16 s, and a final extension at 72 °C for 5 min. Upon completion, 3 μL of each sample was isolated for agarose gel electrophoresis analysis and the Cas12a cleavage assay. The primers used for PCR are listed in Appendix A. All primers were synthesized by TSINGKE (Beijing, China).

### 2.4. Optimize the CRISPR/Cas12a Detection System Using ROX or JOE-Dye ssDNA-FQ Reporters

The CRISPR/Cas12a detection assay mainly contains the Cas12a protein, sgRNA, an ssDNA-FQ reporter, and buffer. The ssDNA-FQ reporter is digested by the Cas12a protein after the formation of the ternary complex (Cas12a protein, sgRNA and target DNA), releasing a fluorescent signal. The volume of the Cas12a cleavage assay was 20 μL, which contained 250 nM Cas12a, 500 nM sgRNA, 300 nM ssDNA-FQ reporter (JOE-N12-BHQ1 or ROX-N12-BHQ2) [31], and 2 μL NEBuffer 2.1. The Cas12a reactions were incubated at 37 °C for 20 min and inactivated at 98 °C for 2 min.

### 2.5. Sensitivity of CRISPR/Cas12a Combined with PCR Assay

The copy number of the PEDV-ORF3 plasmid containing a 320 bp sequence was calculated based on the plasmid, and the insert molecular weight was as follows: number of copies = (amount × 6.022 × 10^23^)/(length × 1 × 10^9^ × 660) (https://toptipbio.com/dna-copy-number-qpcr/, accessed on 20 February 2022). The plasmid DNA was serially diluted from 8 × 10^7^ to 1 × 10^0^ copies/μL. PCR was performed on the dilution series, and the 3 μL products were analyzed by gel electrophoresis. Parallel samples were then analyzed using the Cas12a cleavage reaction, incubated at 37 °C for 15 min and inactivated at 98 °C for 2 min, and the fluorescence was read using a blue or UV Light Transilluminator or a microplate reader.

### 2.6. Optimization of the CRISPR/Cas12a Combined with LAMP Assay (LAMP-Cas12a)

Several sets of LAMP primer pairs were designed based on the highly active sgRNA targeting the PEDV *ORF3* gene (Appendix A), and the PEDV-ORF3 plasmid DNA (10 ng) was then amplified by the LAMP assay. The LAMP amplification assay contained 0.16 U of Bst 3.0 DNA polymerase (NEB, Ipswich, MA, USA), 2.5 μL of 10 × isothermal amplification buffer, 6 mM MgSO_4_, 1.4 mM each of dNTP Mix (Cwbio, CW0941S, CoWin BioSciences, Cambridge, MA, USA), and 2.5 μL of primer mix (10 × primer: 1.6 μM of PEDV-ORF3-FIP/BIP; 0.2 μM of PEDV-ORF3-F3/B3, 0.4 μM of PEDV-ORF3-LF/LB). LAMP reactions were incubated at 65 °C for 40 min and then inactivated at 85 °C for 5 min. The 3 μL products were analyzed by gel electrophoresis or by the Cas12a cleavage reaction. The best primer pairs were then determined based on their fluorescence intensity combined with gel electrophoresis bands to determine the strategy for the LAMP-Cas12a assay.

The range of reaction temperatures (T) or times (t) for the LAMP amplification assay are broad; the temperature range is 50–72 °C, and the time range is greater than or equal to 5 or 10 min. Therefore, different temperatures or times were designed according to the reaction range to determine the reaction conditions for this experiment. First, to determine the reaction temperature, a total of eight temperature (°C) gradients—53, 55, 57.5, 60, 62.5, 65, 67.5, 71—were designed, and the reaction temperature of the negative control was 65 °C. Then, to determine the reaction time, a total of eight reaction times (min)—5, 15, 20, 25, 30, 35, 40, 45—were designed, and the reaction time of the negative control was 45 min. Finally, the analysis of the LAMP amplification products was consistent with the above.

### 2.7. Sensitivity of the CRISPR/Cas12a Combined with LAMP Assay

The copy number of the plasmid DNA was calculated based on the plasmid and insert molecular weight. The plasmid DNA was serially diluted from 1 × 10^4^ to 1 × 10^−1^ copies/μL. LAMP assay was performed on the dilution series, and the 3 μL products were analyzed by gel electrophoresis. Parallel samples were then analyzed using the Cas12a cleavage reaction, incubated at 37 °C for 10 min and inactivated at 98 °C for 2 min, and the fluorescence was read using a blue or UV Light Transilluminator or a microplate reader.

### 2.8. Specificity of the CRISPR/Cas12a Combined with LAMP Assay

First, to evaluate the specificity of the LAMP primer pairs, the four plasmid DNA samples (10 ng) of the diarrhea CoVs were amplified with the following pairs of primers: PEDV-LAMP-2, TGEV-LAMP-1, PDCoV-LAMP-1, and SADS-CoV-LAMP-1 (Appendix A). LAMP reactions were incubated at 65 °C for 20 min and inactivated at 85 °C for 5 min, and the products were verified by agarose gel electrophoresis. Then, to further determine the specificity of the sgRNA of the CRISPR/Cas12a cleavage assay, four amplicons were assayed using the Cas12a detection method with the specific sgRNA (Appendix A).

### 2.9. CRISPR/Cas12a Combined with RT-LAMP Assay

To determine whether the RT-LAMP-Cas12a assay can be used for the detection of RNA viruses, PEDV RNA was amplified using the RT-LAMP assay, and the products were cleaved by Cas12a. To further increase the accuracy of Cas12a, the two systems, RT-LAMP amplification and Cas12a cleaved assay, were integrated into one system. The 25 μL reagent was administered into the bottom of the tube containing the RT-LAMP reaction solution and 12.5 μM ROX-dye ssDNA-FQ reporter. The pipe cap reagent contained 1000 nM of sgRNA, 500 nM of the Cas12a protein, and 2 μL of NEBuffer 2.1. In addition, paraffin oil was added to the liquid surface to prevent evaporation and to control temperature. Reactions were incubated at 65 °C for 40 min and incubated at 37 °C for 15 min after centrifugation to observe the colorimetric/fluorescence readouts.

### 2.10. Detection of PDEV, TGEV, PDCoV, and SADS-CoV by a Multiplex LAMP-Cas12a Assay

To address a coinfection scenario in actual production, a plasmid DNA mixture of PEDV, TGEV, PDCoV, and SADS-CoV (1:1:1:1) was amplified using multiplex LAMP with an optimized primer ratio. The multiplex LAMP reactions were incubated at 65 °C for 20 min and inactivated at 85 °C for 5 min. In addition, paraffin oil was added to the liquid surface to prevent evaporation and to control the temperature. The pipe cap reagent contained 1000 nM of sgRNA, 500 nM of the Cas12a protein, and 2 μL of NEBuffer 2.1. After the multiplex LAMP reaction, the CRISPR/Cas12a solution was centrifuged to observe the colorimetric/fluorescence readout. The amplicons were analyzed using the Cas12a cleavage reaction, incubated at 37 °C for 20 min, and inactivated at 98 °C for 2 min. The co-infected viruses were distinguished by different CRISPR/Cas12a assays.

## 3. Results

### 3.1. Development and Optimization of PEDV, TGEV, PDCoV, and SADS-CoV Detection Based on CRISPR/Cas12a Cleavage Assay

To establish CRISPR-/Cas12a-based nucleic acid detection technology for PEDV, TGEV, PDCoV, and SADS-CoV, we first identified the most active sgRNA to detect each CoVs. We designed seven sgRNAs targeting the PEDV *ORF3* gene (Appendix A), and used the PEDV-ORF3 plasmid as a template for amplification (Appendix A). We then collected the PCR purified products as the DNA template and the complementary ssDNA activator targeted by the sgRNA sequence to test the activity of the seven sgRNAs. The results showed that an sgRNA named sgRNA5 showed the highest activity (Figure 1A and Appendix A). We then identified the sgRNAs with the highest activity against the TGEV *N* gene, the PDCoV *N* gene, and the SADS-CoV *N* gene, which were sgRNA2, sgRNA9, and sgRNA6, respectively (Appendix A and Figure 1B–D).

In addition, the colorimetric signals of CRISPR/Cas12a cleavage assay using JOE-N12-BHQ1 and ROX-N12-BHQ2 reporters were compared. We found that the reaction of using ROX-N12-BHQ2 as a reporter could be clearly observed with directly naked eye (Appendix A). The solution changed visibly as the concentration of the ssDNA-FQ reporter increased (Appendix A), indicating that the selection of ROX-N12-BHQ2 as ssDNA-FQ reporter facilitates CRISPR/Cas12a nucleic acid visual detection, and the optimal concentration was 12.5 µM (Appendix A). To further improve the naked-eye detectability of the colorimetric signal of CRISPR/Cas12a nucleic acid detection, we optimized the concentrations of the sgRNA and the Cas12a enzyme, and the results showed that the optimal concentrations were 1 µM and 0.5 µM, respectively (Appendix A).

### 3.2. Establishment of the CRISPR/Cas12a Combined with LAMP Assay for the Detection of PEDV, TGEV, PDCoV, and SADS-CoV

Next, a CRISPR/Cas12a combined with LAMP assay was established for the detection of four pathogenic swine enteric CoVs. We designed four sets of LAMP primers based on sgRNA5 targeting the PEDV *ORF3* gene, and selected the best primer pairs by gel electrophoresis and the CRISPR/Cas12a cleavage assay. As shown in Appendix A, the LAMP-2 primer pairs for PEDV had the highest amplification efficiency. The temperature range for LAMP amplification was between 57.5 °C and 67.5 °C, and 65 °C was used as the optimal reaction temperature in our study (Appendix A). When the amplification time (t) was ≥15 min (Lane 2), the LAMP products at 65 °C using the LAMP-2 primer pairs could be clearly detected in the CRISPR/Cas12a cleavage assay with the naked eye (Appendix A). However, the most appropriate amplification time was 20 min by agarose gel electrophoresis (Appendix A).

For TGEV, the results showed that the primer pairs LAMP-2, LAMP3, and LAMP-4 caused false-positive readouts after visual detection by CRISPR/Cas12a cleavage assay in the no-template controls (Appendix A). Thus, the best LAMP amplification primer pair for the TGEV *N* gene was LAMP-1 (Appendix A). There was no significant difference in the amplification efficiency of PDCoV. In subsequent experiments, LAMP-1 was selected as the target primer pair for the PDCoV *N* gene (Appendix A). Similarly, LAMP-1 was chosen as the best primer pair for the SADS-CoV *N* gene (Appendix A).

### 3.3. Sensitivity Comparison between CRISPR/Cas12a Combined with PCR and LAMP Assays for the Detection of PEDV, TGEV, PDCoV, and SADS-CoV

We evaluated the sensitivity of the PCR or LAMP-Cas12a assay for the detection of four pathogenic swine enteric CoVs. The PEDV-ORF3 plasmid DNA was diluted from 8 × 10^7^ to 8 × 10^0^ copies/μL for PCR reaction. As shown in Appendix A, the limit of detection (LoD) for the PCR assay was 8 × 10^2^ copies/μL according to the concentration gradient test (Lane 6). Subsequently, the PCR amplification products of ORF3 were further digested by Cas12a cleavage assay and then detected by the blue or UV Light Transilluminator. The results showed that the sensitivity of the PCR-Cas12a assay for PEDV could reach 8 × 10^2^ copies/μL using the plasmid containing *ORF3* as template (tube 6) (Appendix A), but the LoD with the naked eye detection of PEDV by PCR-Cas12a was 8 × 10^3^ copies/μL (Lane 5) (Appendix A). Furthermore, we collected the fluorescence signal values of the colorimetric/fluorescence Cas12a cleavage assay through a multifunctional microplate reader, and the result was the same as those of the blue or UV light assay (Appendix A).

To further evaluate the sensitivity of the LAMP-Cas12a assay, we diluted the plasmid DNAs of PEDV-ORF3, TGEV-N, PDCoV-N, and SADS-CoV-N from 1 × 10^4^ to 1 × 10^−1^ copies/μL. As shown in Figure 2A,C,E,G, the LoD for LAMP-Cas12a with the naked eye almost reached a single copy (tube 6), corresponding with the LoD of the multifunctional microplate reader (Figure 2B,D,F,H), which had a sensitivity a thousand times higher than that of the PCR-Cas12a assay (Appendix A).

### 3.4. Specificity Analysis of the LAMP-Cas12a Assay for the Detection of Four Porcine Coronaviruses

To confirm the specificity of the LAMP-Cas12a assay, we evaluated the specificity of the LAMP primer pairs and the sgRNA. First, LAMP primer pairs targeting the PEDV *ORF3* gene, the TGEV *N* gene, the PDCoV *N* gene, or the SADS-CoV *N* gene were used to amplify the four viral plasmids above. The results showed that each set of primer pairs could only amplify the unique band of the plasmid containing the target gene, indicating that the LAMP primer pairs designed for each virus in this study were specific (Figure 3A). Subsequently, to verify the specificity of the sgRNA, the highly active sgRNAs of the four CoVs were analyzed by the colorimetric/fluorescence Cas12a cleavage assay. As shown in Figure 3B, only the tube containing the corresponding target fragment had colorimetric/fluorescence signals, while the other tubes had no signal readout, Thus, the sgRNA identified in this study is specific.

### 3.5. Detection of Clinical RNA Samples of the Four Porcine Coronaviruses with an RT-LAMP-Cas12a Assay

To apply this newly developed method to field samples, we attempted to detect viral RNA with RT-LAMP-Cas12a directly in a one-tube assay (Figure 4A). As shown in Figure 4B, strong visual colorimetric/fluorescence signals appeared when samples had the corresponding virus. In the no-template control, there was no visual colorimetric/fluorescence signal. Therefore, the method based on CRISPR/Cas12a combined with RT-LAMP assay realized the detection of RNA in clinical samples. In addition, the one-tube operation process has the potential to reduce the aerosol pollution caused by opening the cover, and can also save time. Therefore, one-tube detection will become a useful method for detecting viral infections.

### 3.6. Detection of the Four Porcine Coronaviruses by a Multiplex LAMP-Cas12a Assay

In clinical cases, the symptoms of pig diarrhea are generally caused by cross-infection. Multiplex amplification is able to amplify multiple target genes simultaneously in one reaction, which can greatly improve detection efficiency. In the process of multiplex RT-LAMP amplification, primer pairing and competitive amplification could cause many problems, such as dimerizations, nonspecific amplification, and low amplification efficiency. Therefore, it is necessary to optimize the concentration ratio of the primers for each target sequence. First, we explored the concentration ratio of LAMP primers for amplifying the PEDV *ORF3* gene and the TGEV *N* gene, and established a double-LAMP amplification combined with the CRISPR/Cas12a cleavage assay. The results showed that the PEDV *ORF3* gene and the TGEV *N* gene could be amplified simultaneously in one-tube reaction assay (Appendix A), and lower concentrations (1/2 or 1/3 of the original concentration) of LAMP amplification primers targeting the PEDV-ORF3 gene were beneficial to CRISPR/Cas12a visual detection (Appendix A). This suggested that appropriately reducing the primer concentration could prevent the interaction between primer sequences to a certain extent.

Subsequently, triple-LAMP amplification was established to identify PEDV, TGEV, and PDCoV simultaneously. The results showed that the LAMP amplification primer pairs targeting the TGEV *N* gene had a strong amplification ability, and the primer concentration had little effect on it (Appendix A). However, the LAMP amplification efficiencies of PEDV and PDCoV depended on the primer concentration (Appendix A). Based on the above results, we established quadruple-LAMP amplification (Appendix A). The results showed that the concentration of LAMP primer pairs targeting the TGEV *N* gene had no significant effect on its amplification efficiency, while the amplification efficiencies of the PEDV *ORF3* gene, the PDCoV *N* gene and the SADS-CoV *N* gene were positively correlated with primer concentration (Appendix A). It was suggested that the primer concentrations for PEDV, PDCoV, and SADS-CoV could be appropriately increased, and the primer concentration for TGEV could be reduced to achieve better detection efficiency with multiplex LAMP amplification.

To balance the amplification efficiency of multiplex LAMP and enhance the colorimetric/fluorescence readout of the CRISPR/Cas12a cleavage assay, we adjusted the LAMP primer concentrations (2/5, 6/25, 3/5, and 3/5 of the original concentrations for PEDV, TGEV, PDCoV, and SADS-CoV, respectively). In addition, to avoid aerosol pollution, we established a one-tube multiplex LAMP combined with the CRISPR/Cas12a system (Figure 5A), which was successfully used for the plasmid DNA of the four porcine CoVs (Figure 5B).

## 4. Discussion

To simplify the detection and on-site diagnosis, we developed a multiplex nucleic acid detection method based on CRISPR/Cas12a and multiplex RT-LAMP for the detection of four porcine diarrhea CoVs. Several studies have reported gel-based PCR and qPCR for the simultaneous detection of porcine diarrheal viruses [32]. In comparison, multiplex LAMP-Cas12a assay has the following advantages: when combined with RT-LAMP, complete reverse transcription and amplification can be completed at constant temperature. We found that the LoD for LAMP-Cas12a with the naked eye reached a single copy, which indicated a sensitivity a thousand times higher than that of PCR-Cas12a (from 8 × 10^2^ to 8 × 10^3^ copies/μL) visual detection of the four CoVs. The possible reason is that the amplification and detection efficiency in combination with LAMP is higher than that in PCR. Thus, the multiplex LAMP-Cas12a assay had potentially achieved the sensitivity of a single copy, but further validation was required, while the use of the CRISPR/Cas12a system guaranteed a higher detection specificity, based on Cas12a nuclease and sgRNA detection and typing of target DNA. In addition, as the *ORF3* gene of PEDV has been used in several reports to differentiate between field-and vaccine-derived isolates, it was chosen in this study for the detection of classical PEDV strains [33].

To achieve naked-eye visual detection of viruses, a variety of ssDNA-FQ reporters have been developed [31]. The ROX-labeled ssDNA-FQ reporter allows a direct interpretation of the assay results by observation of visual colorimetric changes with the naked eye [31]. This ssDNA-FQ reporter could be synthesized stably from biological companies in large quantities, in contrast to the expensive gold nanoparticles used in Cas12a/Cas13a-AuNP visual assays [34,35,36,37,38]. The operation process of the one-tube method does not require the transfer of pre-amplified products, which avoids false-positive readouts due to aerosol contamination by pre-amplified product formation [39,40] and saves detection time. For example, the total detection time of the multiplex LAMP-Cas12a assay is only about 25 min.

Compared with other previously reported nucleic acid detection platforms based on CRISPR, which could only detect one pathogen per test [18,19,41], our multiplex LAMP-Cas12a assay could amplify four pathogens with only one-tube reaction, greatly improving diagnosis efficiency. The SHERLOCK V2 platform performed four-channel multiplex detection using different orthogonal cas effectors, and the CARMEN method was able to simultaneously differentiate 169 pathogens in a microwell array [42,43]. However, they both require specialized instruments and expensive experimental reagents, making them difficult to apply to on-site diagnostics. A multiplex RT-RPA combined with separate LbCas12a/AuNP visual assay, reported to simultaneously detect five viruses in apples, has application prospects for on-site deployment in the field [44]. Compared with preamplification by multiplex RT-RPA, our LAMP-Cas12a assay combined with multiplex isothermal amplification assays is cost-effective and more sensitive (almost up to single copy). In fact, it is very difficult to simultaneously detect four CoVs in a single tube, so while our method using only multiplex RT-LAMP is performed simultaneously, it requires different CRISPR/Cas12a assays to distinguish co-infected viruses. In the future, we plan to continue to improve the multiplex LAMP-Cas12a assay in a single tube.

## 5. Conclusions

Taken together, we successfully established a multiplex, field-deployable, and visual naked-eye detection method for the detection of four porcine diarrhea epidemic CoVs in one test. This newly developed viral RNA diagnostic method can help diagnose the status and prevalence of four porcine enteric CoVs.

## Figures and Tables

**Figure 1 viruses-14-00833-f001:**
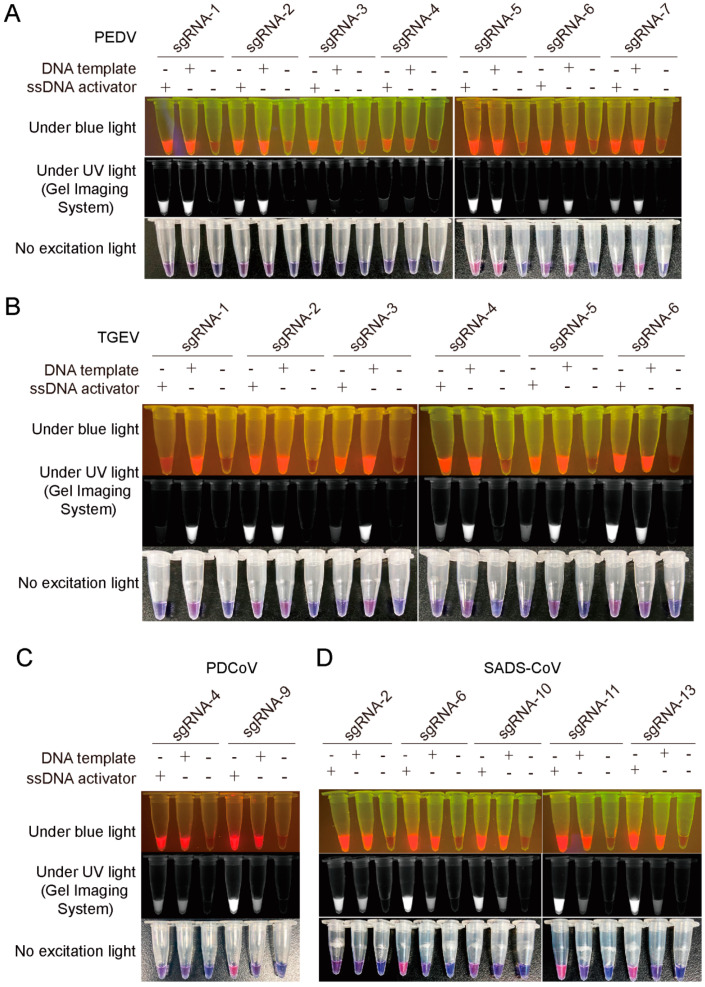
Screening of highly active sgRNAs for detection of PEDV, TGEV, PDCoV, and SADS-CoV. Identification of the highly active sgRNA targeting PEDV *ORF3* gene (**A**), TGEV *N* gene (**B**), PDCoV *N* gene (**C**), and SADS-CoV *N* gene (**D**) by CRISPR/Cas12a cleavage assay; DNA template represents the amplification fragment of *ORF3* or *N* gene as the template; ssDNA activator represents the complementary single-stranded DNA of the sgRNA. No excitation light stands for white light. PEDV, porcine epidemic diarrhea virus; TGEV, transmissible gastroenteritis virus; PDCoV, porcine deltacoronavirus; SADS-CoV, swine acute diarrhea syndrome coronavirus.

**Figure 2 viruses-14-00833-f002:**
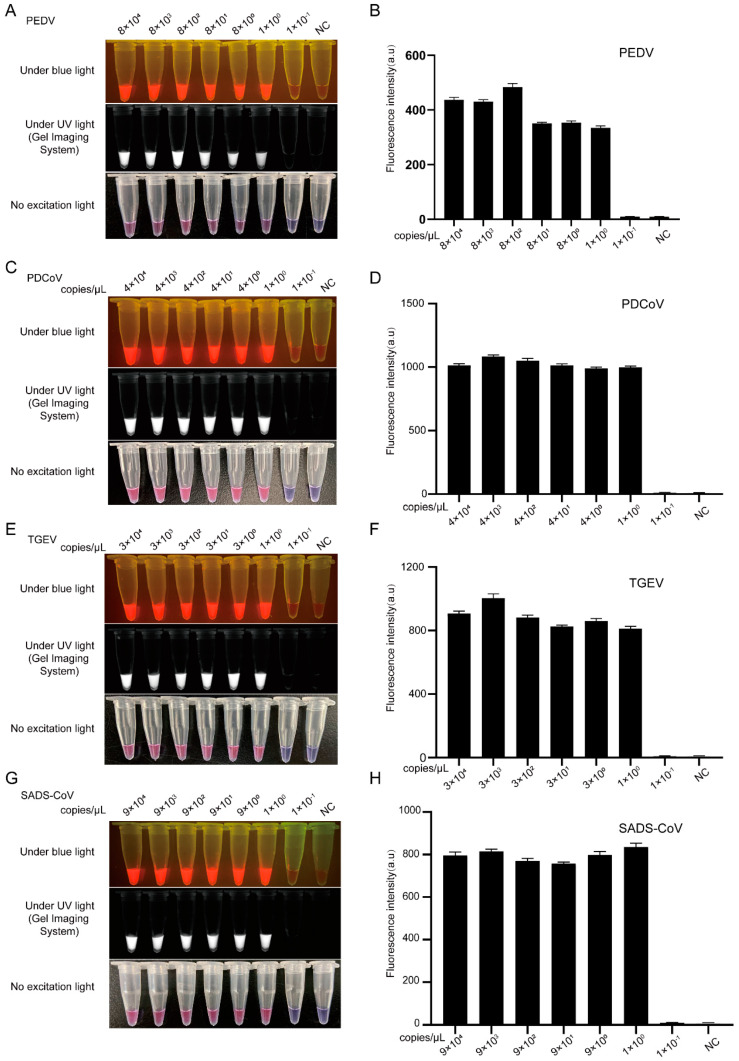
Sensitivity of CRISPR/Cas12a combined with LAMP assay for portable detection of PEDV, PDCoV, TGEV, and SADS-CoV. (**A**,**C**,**E**,**G**) Colorimetric/fluorescence signals from a series of 10-fold dilutions of plasmid DNA by CRISPR/Cas12a coupled with LAMP assay. (**B**,**D**,**F**,**H**) Sensitivity of the CRISPR/Cas12a coupled with LAMP assay quantified by a multi-functional microplate reader (*n* = 3). No excitation light stands for white light; PEDV, porcine epidemic diarrhea virus; TGEV, transmissible gastroenteritis virus; PDCoV, porcine deltacoronavirus; SADS-CoV, swine acute diarrhea syndrome coronavirus; NC, negative control. Data are represented as mean ± SEM.

**Figure 3 viruses-14-00833-f003:**
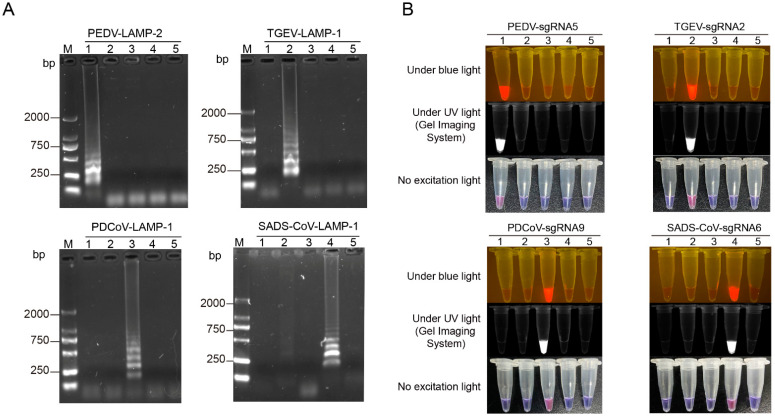
Specificity of LAMP-Cas12a assay for detection of four porcine coronaviruses. (**A**) The agarose gel electrophoresis for detection of specific LAMP amplicons; Lane M: 2000 DNA Ladder; bp: base pairs; Lane 1: PEDV-ORF3 plasmid as DNA template; Lane 2: TGEV-N plasmid as DNA template; Lane 3: PDCoV-N plasmid as DNA template; Lane 4: SADS-CoV-N plasmid as DNA template; Lane 5: non-template control (NTC). (**B**) The colorimetric/fluorescence signal detection under the blue (470 nM) and UV light; LAMP-amplified products of PEDV-ORF3 plasmid (1), TGEV-N plasmid (2), PDCoV-N plasmid (3), SADS-CoV-N plasmid (4), non-template control (5). No excitation light stands for white light; PEDV, porcine epidemic diarrhea virus; TGEV, transmissible gastroenteritis virus; PDCoV, porcine deltacoronavirus; SADS-CoV, swine acute diarrhea syndrome coronavirus.

**Figure 4 viruses-14-00833-f004:**
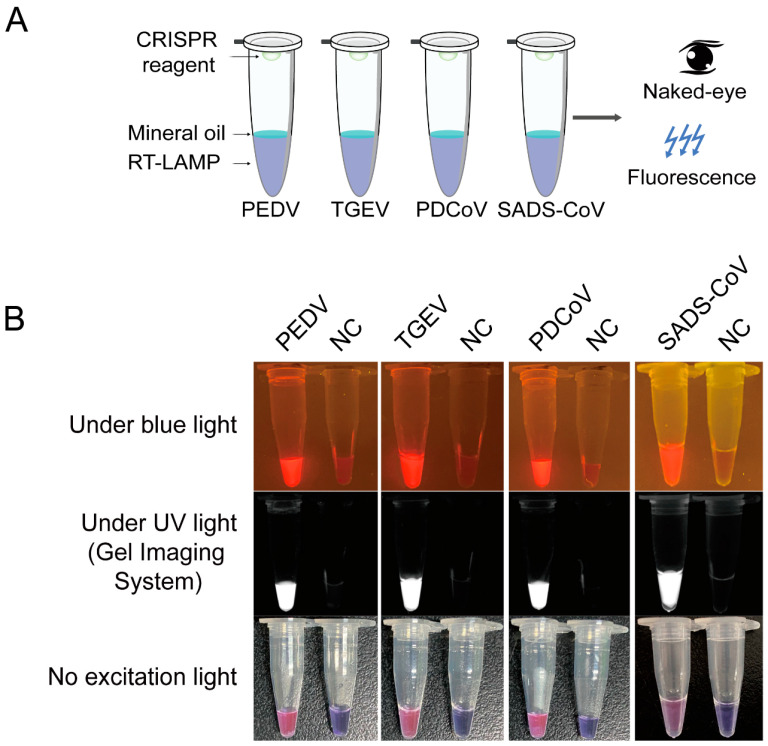
One-tube detection of four porcine coronaviruses in clinical samples based on CRISPR/Cas12a combined with RT-LAMP assay. (**A**) Diagram of principle of the contamination-free one-tube CRISPR/Cas12a combined with RT-LAMP assay detection of clinical samples. (**B**) Colorimetric/fluorescence readout of four porcine CoVs by one-tube multiplex RT-LAMP-Cas12a assay. CRISPR, clustered regularly interspaced short palindromic repeats; RT-LAMP, reverse transcription-loop-mediated isothermal amplification; No excitation light stands for white light; PEDV, porcine epidemic diarrhea virus; TGEV, transmissible gastroenteritis virus; PDCoV, porcine deltacoronavirus; SADS-CoV, swine acute diarrhea syndrome coronavirus; NC, negative control.

**Figure 5 viruses-14-00833-f005:**
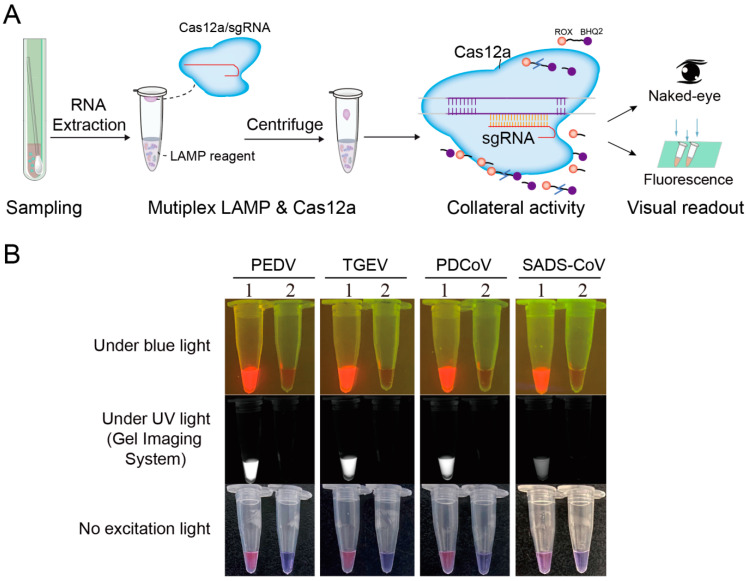
Establishment of one-tube multiplex LAMP-Cas12a assay for detection of four porcine diarrhea coronaviruses. (**A**) Schematic diagram experimental flow and principle of contamination-free one-tube multiplex LAMP-Cas12a assay. (**B**) Colorimetric/fluorescence readout of four porcine coronaviruses by one-tube multiplex LAMP-Cas12a assay. 1: plasmid DNA mixture of PEDV, TGEV, PDCoV, and SADS-CoV (1:1:1:1) as templates; 2: non-template control (NTC). sgRNA, small guide RNA; LAMP, loop-mediated isothermal amplification; ROX-BHQ2, ROX-labeled ssDNA-FQ reporter; No excitation light stands for white light; PEDV, porcine epidemic diarrhea virus; TGEV, transmissible gastroenteritis virus; PDCoV, porcine deltacoronavirus; SADS-CoV, swine acute diarrhea syndrome coronavirus.

## Data Availability

The data presented in this study are available on request from the corresponding author.

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
