# Peer review of "Detection of Four Porcine Enteric Coronaviruses Using CRISPR-Cas12a Combined with Multiplex Reverse Transcriptase Loop-Mediated Isothermal Amplification Assay"

_viruses, 2022, doi:10.3390/v14040833_

Round 1

Reviewer 1 Report

Authors of the current manuscript describe a multiplex CRISPR-Cas12a-based assay to detect four porcine enteric coronaviruses.

In the introduction section, authors fail to provide strong motivation for the need for multiplex detection of the four porcine viruses using naked eye. Additionally, there isn't any schematic or description of the assay itself to help the readers understand the assay principle. 

In the study design it was not very clear as to how many times any of the conditions were repeated to assess reproducibility of the results. In the final study design of multiplex detection, authors don't discuss the sensitivity of the assay. 

Author Response

Point 1:

In the introduction section, authors fail to provide strong motivation for the need for multiplex detection of the four porcine viruses using naked eye.

Response 1:

Respected reviewer, thank you very much for your valuable suggestions. Your suggestions made a huge improvement in our research paper. According to your comment, this point has been clarified in the revised paper. As described in lines 58-61 of the revised version, “Therefore, it is urgent to develop a POC diagnostic platform for rapidly detecting multiple porcine diarrhea CoVs in the field, which can be quickly applied to diagnosis in resource-poor or remote small pig farms. Especially for the simultaneous on-site measurement of different CoVs from a single sample, called multiplexed POC testing is necessary”.

Point 2:

Additionally, there isn't any schematic or description of the assay itself to help the readers understand the assay principle.

Response 2:

Thank you for pointing this out. According to your suggestion, as can be seen in Figure 5A (revised version), the schematic of the assay has been updated to help the readers easily understand the assay principle.

Point 3:

In the study design it was not very clear as to how many times any of the conditions were repeated to assess reproducibility of the results.

Response 3:

We feel sorry that we did not provide enough information about the number of repetitions in our study. All CRISPR/Cas12a combined with LAMP assays were repeated at least twice. And the sensitivity of the LAMPCas12a assay semi-quantified by a multi-functional microplate reader was performed 3 times (Figure 2).

Point 4:

In the final study design of multiplex detection, authors don't discuss the sensitivity of the assay.

Response 4:

According to your suggestion, the sensitivity of the assay was discussed. As described in lines 340-344 of the revised version, “We found that the LoD for LAMP-Cas12a with the naked eye reached a single copy, which had a thousand times higher sensitivity than that of PCR-Cas12a (8 × 102 - 8 × 103 copies/μL) visual detection of four CoVs. The reason possible is that the amplification and detection efficiency in combination with LAMP is higher than that in PCR. Therefore, multiplex LAMP-Cas12a assay have the potential achieved the sensitivity of a single copy, but further validation is required”.

Reviewer 2 Report

In this manuscript, authors combined RT-LAMP specific amplification and CRISPR/Cas12a-cleaved FQ-reporter to develop a multiplex, rapid, sensitive, on-site molecular differential diagnosis technology for porcine enteric coronaviruses. It is a novel strategy for detection of swine enteric coronaviruses with reference value. Therefore, I recommend posting it on Viruses. But there is a small part of the text and figure legends that need to be corrected or modified.

  1. The statements on lines 77-78 and lines 374-375 are not appropriate. Here authors claimed that their technology can be used to rapid and efficiently detect four pathogenic swine enteric CoVs “simultaneously”. In fact, only multiplex RT-LAMP is performed simultaneously, and co-infected viruses still need to be distinguished by different CRISPR/Cas12a assays. But this statement is only appeared in the last sentence of the Materials and Methods section (lines 193-194).
  2. Figure S5, what viral DNA was used as template here, PDCoV-N or TGEV-N? Figure S1 and S2, the word "different" is all mistyped here. Figure S8, the word "fluorescence" is all mistyped here. It is recommended that authors can check the description of all the figures again.

Author Response

Point 1:

The statements on lines 77-78 and lines 374-375 are not appropriate. Here authors claimed that their technology can be used to rapid and efficiently detect four pathogenic swine enteric CoVs “simultaneously”. In fact, only multiplex RT-LAMP is performed simultaneously, and co-infected viruses still need to be distinguished by different CRISPR/Cas12a assays. But this statement is only appeared in the last sentence of the Materials and Methods section (lines 193-194).

Response 1:

Thanks for your suggestions. We feel sorry for the improper wording. This has now been corrected. The title of the manuscript has been changed to “Detection of four porcine enteric coronaviruses using CRISPR-Cas12a combined with multiplex reverse transcriptase loop-mediated isothermal amplification assay”. In this regard, we provided a brief discussion of the manuscript. In order to enlarge this section, we added the following information: (lines 364-366) “In fact, it is very difficult to simultaneously detect four CoVs in a single tube, so our method using only multiplex RT-LAMP is performed simultaneously and still requires different CRISPR/Cas12a assays to distinguish co-infected viruses. In the future, we plan to continue to improve the multiplex LAMP-Cas12a assay in a single tube”.

Point 2:

Figure S5, what viral DNA was used as template here, PDCoV-N or TGEV-N? Figure S1 and S2, the word "different" is all mistyped here. Figure S8, the word "fluorescence" is all mistyped here. It is recommended that authors can check the description of all the figures again.

Response 2:

Thanks for your careful checks. We are sorry for our carelessness. Thanks for your correction. It was indeed a serious grammatical error. And we have corrected it according to your suggestion. In Figure S5, PDCoV-N plasmid DNA was used as template, we have make clarified in the revised figure legend.

Reviewer 3 Report

Overall Comments

Up to now, PCR-based diagnostics still have many challenges including time & labor-consuming, expensive equipment needed and cross contamination, for the detection of porcine diarrhea coronaviruses. Therefore, there is an urgent need to develop a time and cost saving and efficient diagnostic platform for rapid detection of the pig diarrhea coronaviors.

The main purpose of this paper is to develop and verified a point-of-care diagnostic platform named as reverse transcriptase loop-mediated isothermal amplification (RT-LAMP) for rapidly detecting multiple porcine diarrhea coronaviruses. On this propose Liu and co-authors created an efficient CRISPR-Cas12a based detection method which can be used to rapid and efficiently detect four pathogenic swine enteric CoVs (TGEV, PEDV, SADS-CoV and PDCoV) simultaneously, thus greatly improved the diagnosis efficiency. Compared with preamplification by multiplex RT-RPA, this method using multiplex RT-LAMP has many advantages include sensitive, specific without cross-reaction, cost-effective, field-deployable and naked eye visible without professional instrument, which is potential of further popularization and application.

Overall, the manuscript is well organized, the methods described in good detail, and the figures with corresponding legends provide the data in a clear form. The conclusions of the study are supported by the data presented, and are clearly stated. Reviewer believe that this manuscript with interesting information for readers involved in the swine enteric CoVs detection methods.

Before deciding whether to publish this article, the reviewer is concern about the detection results about PEDV, which the author designed the LAMP primer based on the active sgRNA targeting the PEDV ORF3 gene, instead of using the N gene as the target gene for detection like the TGEV, SADS-CoV and PDCoV. To the reviewer’s knowledge, ORF3 gene usually has large fragment deletion or truncation in PEDV field strains or cell adapted strains. Recently researchers also found that this truncation in ORF3 gene was naturally occurring (doi.org/10.3390/v13081562; doi.org/10.3390/v14030487; doi.org/10.1007/s11262-007-0164-2). Therefore, using ORF3 gene as target gene has the risk of missing detection. In view of this reason, it is hard to tell if some of my concerns relate to data on hand but not shown, but from what I can see here, some of my requests may require additional experiments, which will increase the scientific significance of the manuscript.

Major comment-

Why you choose the ORF3 gene instead of N gene, as the target gene for PEDV detection.  If you have the experiment results about N gene, please incorporate into the manuscript. If not, I suggest that you should perform the extra experiment to prove this or delete the PEDV detection results throughout this manuscript.

Minor comments-

Line 127, the length of ORF3 gene in the plasmid is 320bp, so correct the 265 to 320.

Supplementary Information: All the figures in this part is not visible.

Author Response

Point 1:

Why you choose the ORF3 gene instead of N gene, as the target gene for PEDV detection. If you have the experiment results about N gene, please incorporate into the manuscript. If not, I suggest that you should perform the extra experiment to prove this or delete the PEDV detection results throughout this manuscript.

Response 1

We feel sorry that we did not provide enough information about choose the ORF3 gene instead of N gene for PEDV detection. This was described in Materials and methods section (Line 81), The ORF3 gene has been used in several reports to differentiate between field-and vaccine-derived isolates, in this study, ORF3 gene was chosen for detection of classical PEDV strains. We added the following information: (lines 345-347) “In addition, the ORF3 gene of PEDV has been used in several reports to differentiate between field-and vaccine-derived isolates, in this study, ORF3 gene was chosen for detection of classical PEDV strains [33]”.

Point 2:

Line 127, the length of ORF3 gene in the plasmid is 320bp, so correct the 265 to 320.

Response 2

Thanks for your careful checks. And we have corrected it according to your suggestion.

Point 3:

Supplementary Information: All the figures in this part is not visible.

Response 3

According to your suggestion, we provided a PDF version of supplementary materials for review.

Round 2

Reviewer 1 Report

Thank you for addressing the comments and revising the manuscript accordingly.

Reviewer 3 Report

The authors have effectively addressed my concern to fill in the gaps, and I recommend the publication of this manuscript in its present form.